# Relationship between the evaluation of agricultural scientific and technological innovation capacity and the influencing factors of green agriculture

**Mei Zhang[1]\*, Kai Fang[1], Danting Zhang[2,3], Dejie Zeng[1]**

**1** College of Economy and Trade, Zhongkai Univercity of Agriculture and Engineering, Guangzhou, Guangdong, China, **2** Institute of Agricultural Economics and Information, Guangdong Academy of Agricultural Sciences, Guangzhou, Guangdong, China, **3** Office, Guanggdong Institute for Rural Studies, Guangzhou, Guangdong, China

\* 2016121489@jou.edu.cn

## Abstract

China has limited arable land area, and its output value is increased with enhanced agricultural inputs such as machinery, irrigation, fertilizers, and pesticides. However, this mode is accompanied by an increase in agricultural carbon emissions. With the aim to further examine the driving effect of scientific and technological innovation on economic growth and green agriculture, this study uses the Solow growth model coupled with the Cobb–Douglas production function and vector autoregressive models. Then, the agricultural scientific and technological innovation capacity in Guangdong Province during 2006–2020 is evaluated by using the contribution rate of agricultural scientific and technological progress (ASTP) as the assessment index. In addition, the carbon footprints of green agricultural indexes such as machinery, irrigation, fertilizers, and pesticides are measured to analyze the relationship between green agriculture and agricultural scientific and technological innovation capacity. Results demonstrate the gradual increase in the contribution rate of ASTP in Guangdong Province. During the 11th, 12th, and 13th Five-Year Plan periods, the rates were 65.09%, 65.94%, and 70.40%, respectively, indicating that the agricultural scientific and technological innovation ability constantly improved. Among the indexes of green agriculture, the carbon footprints of machinery have a significant impact on agricultural scientific and technological innovation, which is quickly transformed into machinery. Such innovation requires the driving force of science and technology itself, which have relatively significant and rapid effects. On the basis of the results, corresponding policy suggestions are proposed: increasing investments in scientific and technological innovation in the agricultural field, vigorously developing new energy-saving and emission reduction products and processes for fertilizers, and increasing the research and promotion of agricultural machinery. The proposed method provides good prospects for the development of agricultural production towards mechanization, intelligence, efficiency, and greenness.

**Data Availability Statement:** All data files are available from the Guangdong statistical Yearbook database, http://stats.gd.gov.cn/gdtjnj/index.html.

**Funding:** The authors received no specific funding for this work.

**Competing interests:** The authors have declared that no competing interests exist.

# 1. Introduction

Agricultural scientific and technological innovation is a key link and support in implementing the national strategy of promoting agriculture through science and technology as well as its green development of agriculture [1]. To achieve rapid economic development, more attention should be paid on the quality of agicutural development [2]. At present, China has limited arable land area, and its output can be increased with enhanced agricultural inputs such as machinery, irrigation, fertilizers, and pesticides. However, this mode is accompanied by an increase in agricultural carbon emissions. The fundamental way for green and sustainable agricultural development lies in the improvement of quality and production through scientific and technological progress [3, 4].

With the deepening of relevant scientific research in China and the breakthroughs in the contribution rate of ASTP, agricultural production has changed from the traditional labor mode to mechanization and intelligence while the labor production efficiency has continuously improved [5, 6]. According to the public information of the Ministry of Agriculture and Rural Affairs in 2022, the contribution rate of agricultural science and technology progress in the country has exceeded 61%, and the overall research and development strength of agricultural science and technology has entered the forefront of the world, however, several problems remain, such as insufficient R&D funds, low popularization rate of agricultural science and technology, and the disconnection between R&D achievements and the market [7]. Therefore, the internal relationship of promoting the development of green agriculture through the progress of agricultural science and technology is not clear, and its correlation logic has not been clearly explored.

In recent years, human over-exploitation of various natural resources has led to rising carbon emissions, forming a "greenhouse effect" [8]. With the increase of global farmland, agricultural mechanization has been improved and the input in agricultural goods and materials has been increased, but are accompanied by a gradual increase in the proportion of carbon emissions from agricultural production [9]. As revealed by the IPCC, agricultural production in China is the second highest among global carbon emission industries and accounts for approximately 25% of total carbon emissions [10]. The concept of carbon footprint, which evolves from the theory of ecological footprint, is mainly used to evaluate the pressure of human activities on natural resources in the whole life process [11]. Since the concept of carbon footprint was put forward, many Chinese and foreign researchers have extensively explored the influencing factors of agricultural carbon emissions. The research has focused on the assessment of carbon footprints. For example, West explored the concept and index system of carbon footprints by measuring the effect of different agricultural production modes on carbon emissions, and established a primary evaluation system of agricultural carbon emissions [12]. Carbon emissions are huge in actual agricultural production. Ma et al. found that the annual net carbon emissions of rice and wheat planting under the rotation mode in China are 8.36 tons of ce/ha (ce is the equivalent amount of $CO_2$) [13]. In addition, annual $CO_2$ emissions induced by straw burning reach 229 million tons, accounting for 3.8% of the total greenhouse gas emissions [14]. China is a large country in which agriculture is one of the three major industries, and agricultural greenhouse gas emissions account for 21% of the total [15]. Agricultural carbon emissions have a considerable impact on China's overall carbon emissions, and thereby in achieving its "double-carbon" (carbon peak and neutral) goal. Several scholars have probed into the influencing factors of carbon footprint. For example, Wu X R and Zhang J B used the improved Divisia index decomposition method to analyze the decoupling characteristics of agricultural carbon emission drive and emission reduction in China, and thus exhibited their correlation [16]. Given the huge overall agricultural carbon emissions,

specific steps are needed to optimize the modes and links [17]. Laura et al. evaluated the energy efficiency of corn cultivation technology under different farming modes based on the carbon footprint theory, and found that fuel consumption is the largest contributor in the traditional deep-ploughing link with high optimization and promotion potential [18]. At present, farmland carbon emissions are high, and its potential of emission reduction is huge. Relying on scientific and technological progress to save energy and reduce emissions has an important impact on the sustainable green development of agriculture in China. Therefore, exploring the agricultural scientific and technological progress and innovation ability and the relationship between it and green agriculture has strong practical significance for future development and policy making. With reference to the above research methods and results, the present study integrated macro-analysis, specific links, and correlation coupling to explore the relationship between agricultural scientific and technological innovation capacity, green agriculture, and its influencing factors.

The extreme weather expreienced in recent years made world focus on the mode of low-carbon economic development [19], and concern about climate chinaage and its link to energy is increasingly present [20]. Low-carbon transformation has been put forword to improve litimited resoures and vulnerable environmental conditions because of global energy shortages [21]. At the international level, according to the "US–China Joint Announcement on Climate Change", China plans to achieve peak carbon dioxide emissions in 2030 and carbon neutrality in 2060, which, however, have been seriously impeded by increasing agricultural investments in fertilizers, pesticides, and other machineries [22]. In agriculture, scientific and technological progress and the increase in output value have an important impact on carbon emissions. For example, the reduction and controlled release of chemical fertilizers and pesticides, and the high efficiency of agricultural machinery operations resulting from scientific and technological progress can considerably reduce carbon emissions [23]. Green innocation would promote human well-being [24] At present, therefore, the better approach is to fully apply scientific and technological innovation technology to the agricultural field, transform the high-quality achievements of energy conservation and emission reduction, and promote the green and sustainable agricultural development. In this study, when selecting the evaluation index, the agricultural scientific and technological innovation capacity in Guangdong Province is comprehensively analyzed. Meanwhile, from the perspective of green agriculture, the scientific and technological progress, increase in output value, carbon emissions, and their relationships and influencing factors in Guangdong Province are investigated to provide reasonable suggestions for reducing agricultural carbon emissions and developing green and sustainable agriculture.

The theoretical contributions of this study are mainly manifested in the following aspects: First, the agricultural scientific and technological innovation capacity as a comprehensive data of regional agricultural development is evaluated by introducing its contribution rate as an evaluation index. Through the integrated analysis of related data, the annual and stage-by-stage changes of agricultural scientific and technological innovation capacity can be clearly reflected. Second, green agriculture is the general direction of regional development. By introducing carbon footprint as an index, various agricultural input producers can be compared and analyzed in a unified dimension. Different parameter standards corresponding to each agricultural input can also be used to objectively evaluate the correlations and related changes among green agricultural indexes. Third, the development relationship between green agriculture and scientific and technological innovation in Guangdong Province is analyzed, and the applied vector autoregressive (VAR) model is introduced. Each endogenous variable in the system is regarded as the lag value of all such variables, and the contribution rate of ASTP and the carbon footprint of various agricultural inputs are appropriately analyzed to reveal their relationship with green agriculture.

The remainder of this paper is organized as follows: Section 2 presents a literature review, including documents regarding the influence of scientific and technological progress on agricultural development. Section 3 describes the research methods and data on agricultural scientific and technological innovation capacity. For the analysis of the correlation between green agriculture and scientific and technological innovation via a VAR model, the research objects are selected and data sources are introduced. Section 4 presents the empirical study using the VAR model, relationship between green agriculture and agricultural scientific and technological development, and the research results. In Section 5, the research conclusions, policy suggestions, and expectations are discussed.

## 2. Literature review

Previous studies have extensively explored the influence of scientific and technological progress on agricultural development, however, research on scientific and technological progress on the development of green agriculture is not much. Most of the early research focused on the levels of agricultural scientific and technological innovation theory and system construction. For instance, Astrida Miceikiene et al identified the factors affectiong environmental pollution agriculture [25]. Mittler considered agricultural scientific and technological innovation as the role played by science and technology in agricultural construction base [26], Asfaw and Shiferaw found that improved agricultural technology has a significant positive impact on agricultural production [27], and Qin C J and Zhang Z H calculated the contribution rate of Guangdong's scientific and technological progress, and put forward policy suggestions for future agricultural economic growth [28]. In recent years, agricultural scientific and technological innovation has been increasingly and deeply investigated. Tang Y F et al. calculated the contribution rate of ASTP in 14 prefectures of Gansu Province during the Twelfth Five-Year Plan period by using cluster analysis [29]. Huang Y et al. established a Solow growth rate model to measure and analyze the ASTP in Fujian and Taiwan [30]. Considering the present situation and problems of agricultural scientific and technological development in Jiangsu Province, Fu G Q and Qiu S put forward countermeasures and suggestions to support high-quality sustainable agricultural development based on agricultural scientific and technological innovation from multiple subjects and fields [31]. Alessandro Magrini et al put forward a composite indicator of agricultural sustainability in the European Union [32]. Generally speaking, most of these previous studies analyzed the impact of scientific and technological innovation on agricultural production and income increase, but paid little attention to the relationship between scientific and technological innovation and green sustainable agricultural development. Specifically, research is lacking in evaluating the relationship between agricultural scientific and technological innovation capacity and the influencing factors of green agriculture.

In the present study, therefore, the relationship between the evaluation of agricultural scientific and technological innovation capacity and the influencing factors of green agriculture is innovatively proposed. Guangdong Province is taken as an example, to provide a data basis for China's scientific and technological progress and the implementation of the "double-carbon" strategy. The contribution rate of ASTP is an important index to measure the application of scientific innovation in agriculture. This parameter is then selected to evaluate the agricultural scientific and technological innovation capacity by coupling the Solow residual method (also known as the Solow growth rate model) with the Cobb–Douglas production function (C–D function). The Solow growth rate model was established by American economist Solow in 1957 [33, 34]. On this basis, Zhu X G put forward their measurement method for the contribution rate of ASTP, and became the most recognized and widely used in China [35]. Scientific and technological progress exerts influence on the change of agricultural carbon footprints.

The application of advanced scientific and technological innovation in agricultural production is conducive to the sustainable development of green agriculture. Lu N et al researched and discussed the positive impact of agricultural environmental technical efficiency on the growth of green total factor productivity [36], He X X et al reported that Guangdong's agricultural scientific and technological innovation significantly promotes the improvement of agricultural green total factor productivity [37]. Wu F analyzed the regional differences in agroecological efficiency, and discussed the influence of agricultural scientific and technological innovation level on agroecological efficiency [38]. Sun D X and Cheng S T determined that agricultural green technology innovation is the key factor to realize carbon emission reduction [39], Chen F F found that, at present, China has low transformation rate of green technology innovation achievements, and the effect of green energy saving and emission reduction in agriculture is highly difficult to achieve [40]. Mao S P and Yang Y L classified and quantified the agricultural scientific and technological innovation policies issued at the national level since the reform and opening up (1978–2015) from three dimensions, and quantitatively explored its evolution trend and characteristics [41]. Based on the previous research on the progress of agricultural science and technology and green agriculture, this study plans to further deepen the logical relationship. A review of relevant literature also revealed that agricultural production factors such as pesticides, fertilizers, and agricultural machinery are highly influenced by the level of science and technology [42]. In the present study, therefore, the contribution rate of ASTP can be appropriately used to measure agricultural scientific and technological innovation capacity. In addition, green agriculture can be evaluated using the carbon emissions of such agricultural production factors as pesticides, fertilizers, and agricultural machinery to evaluate their correlations. Moreover, each endogenous variable in the system can serve as the lag value of all such variables via the VAR model. Thus, the contribution rate of ASTP can be properly associated with carbon emissions.

## 3. Research methods and data

### 3.1 Evaluation of agricultural scientific and technological innovation capacity

In this study, the contribution rate of ASTP was used to evaluate the agricultural scientific and technological innovation capacity. The basic principle is to assume that agricultural economic growth consists of two parts: one is the increase of production input, that is, the increase of land, labor, and material costs; the rest comes from the increase of the input–output ratio caused by scientific and technological progress [3]. Thus, the agricultural production function model is:

$$Y = AK^{\alpha}L^{\beta}M^{\gamma}e^{\delta t}, \tag{1}$$

where Y, A, K, L, and M are the total agricultural output value, constant, capital input, labor input, and cultivated land input, respectively. $\alpha$, $\beta$, and $\gamma$ are the elastic coefficients of capital, labor, and surface cultivated land, respectively; $\delta$ is the rate of scientific and technological progress; and $t$ is a time variable. Logarithms are simultaneously taken from two sides to obtain:

$$\ln Y = \ln A + \alpha \ln K + \beta \ln L + \gamma \ln M + \delta t. \tag{2}$$

Taking partial derivatives from time 7 can yield:

$$\frac{1}{Y}\frac{d_Y}{d_t} = \alpha\frac{1}{K}\frac{d_K}{d_t} + \beta\frac{1}{L}\frac{d_L}{d_t} + \gamma\frac{1}{M}\frac{d_M}{d_t} + \delta. \tag{3}$$

In a specific year, $d_t = 1$, and the formula is adjusted as follows:

$$\frac{\triangle Y}{Y} = \alpha \frac{\triangle K}{K} + \beta \frac{\triangle L}{L} + \gamma \frac{\triangle M}{M} + \delta. \tag{4}$$

Then, the calculation formula for the scientific and technological progress rate is:

$$\delta = \frac{\triangle Y}{Y} - (\alpha \frac{\triangle K}{K} + \beta \frac{\triangle L}{L} + \gamma \frac{\triangle M}{M}). \tag{5}$$

The contribution rate of ASTP can then be solved by the ratio of the scientific and technological progress rate to the agricultural growth rate:

$$\eta = \delta \Big/ {\textstyle\frac{\triangle Y}{Y}}. \tag{6}$$

The sum of elastic coefficients is assumed to be 1, that is, $\alpha+\beta+\gamma = 1$. Given that cultivated land changes little, its elastic coefficient is taken as 0.25 by reference to the national index and is calculated using the following formula:

$$\alpha_i = \alpha \ln[e - 1 + (\frac{K_0}{L_0} + \frac{K_t}{L_t})/(\frac{K_{0i}}{L_{0i}} + \frac{K_{ti}}{L_{ti}})]. \tag{7}$$

## 3.2 Correlation analysis between green agriculture and scientific and technological innovation based on VAR model

In this section, the carbon footprint of each index (e.g., machinery, irrigation, fertilizers, and pesticide) which influences green agriculture most was calculated and the influence of scientific and technological innovation capacity on green agricultural development was analyzed based on the VAR model. Table 1 shows the carbon footprints induced by the utilization of diesel, irrigation, fertilizers, and pesticides as calculated by their respective indexes [3].

The relationship between green agriculture and scientific and technological innovation in Guangdong Province was then analyzed by using the VAR model. The VAR model is an econometric analysis tool proposed by Sims in 1980 and regards each endogenous variable in the system as a function of the lag value of all such variables, expressed as follows:

$$y_t = A_1 y_{t-1} + \ldots + A_p y_{t-p} + Bx_t + \varepsilon_t, t = 1, 2, \ldots, T. \tag{8}$$

where $\varepsilon_t$ is the white noise sequence vector, $p$ is the autoregressive lag order, and $T$ is the number of samples. The VAR model needs to test the stability of variables, and the analysis of variance and of impulse response function can be implemented only after determining the optimal lag order. In this study, the VAR model simulation was performed using Eviews software. With 2006 to 2021 as the time period, the development relationship between green agriculture and scientific and technological innovation in Guangdong Province was analyzed.

## 3.3 Research data

All data were derived from the Guangdong Statistical Yearbook, including agricultural total output value, agricultural labor force population, cultivated land area, agricultural diesel oil

**Table 1. Carbon footprint coefficient of each index.** http://stats.gd.gov.cn/gdtjnj/index.html.

| Index | Carbon footprint coefficient |
|---|---|
| Diesel | 0.593 kg/kg |
| Irrigation | 25.000 kg/hm$^2$ |
| Fertilizer | 0.896 kg/kg |
| Pesticide | 4.934 kg/kg |

consumption, pesticide application, chemical fertilizer application, and agricultural added value. The statistical yearbook of Guangdong Province was updated until 2020, and thus this study reflects the data between 2006 and 2020. The national major strategic stages were divided into the 11th, 12th, and 13th Five-year Plans, and 5-year overall changes were taken as evaluation nodes. The Five-Year Plan, the full name is the Outline of the Five-Year Plan for National Economic and Social Development of the People's Republic of China, is an important part of China's national economic plan. Looking back on the history of the five-year Plan, we can explore the law of China's economic development. In the field of agriculture, the broad sense of agriculture includes agriculture (plantation industry), forestry, animal husbandry, sideline industry, fishery (aquaculture industry), in this paper, the main study is the narrow sense of agriculture, namely plantation industry.

In accordance with the principle of administrative division of Guangdong Province, Guangdong Province is divided into four parts: Pearl River Delta region (Guangzhou, Shenzhen, Dongguan, Foshan, Zhongshan, Zhuhai, Jiangmen, Huizhou, and Zhaoqing), East wing region (Chaozhou, Shantou, Jieyang, and Shanwei), west wing region (Zhanjiang, Maoming, and Yangjiang) and mountainous region (Shaoguan, Qingyuan, Heyuan, Meizhou, and Yunfu).

## 4. Results analysis

### 4.1 Agricultural scientific and technological innovation in Guangdong Province

Based on the statistical data of Guangdong Province from 2006 to 2020, the contribution rate of ASTP was used to characterize the agricultural scientific and technological innovation capacity in Guangdong Province, as shown in Fig 1 below. The results show fluctuations in the annual growth rate of total agricultural output value and the contribution rate of ASTP in Guangdong Province, while the total contribution rate of ASTP was on the rise. This finding shows that the total agricultural output value of Guangdong Province is increasing, same as the contribution of scientific and technological innovation to the total agricultural output

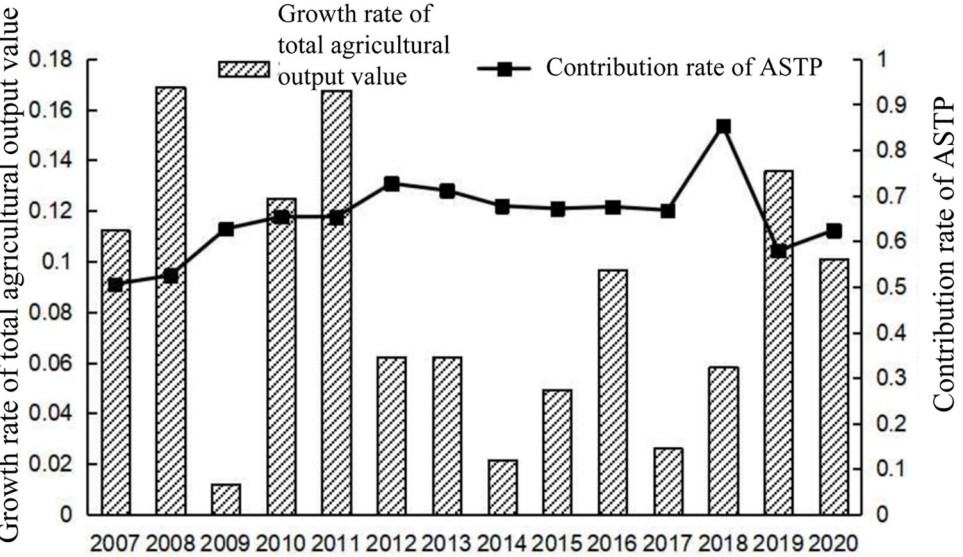

**Fig 1. Year-by-year growth rates of total agricultural output values and contribution rate of ASTP in Guangdong Province from 2007 to 2020.** http://stats.gd.gov.cn/gdtjnj/index.html.

value. Meanwhile, comparing the annual growth of these two variables, the driving effect of scientific and technological innovation on agricultural development is lagging behind. From 2007 to 2012, for example, the growth rate of Guangdong's total agricultural output value reached two peaks in 2008 and 2011 with 16.90% and 16.77%, respectively, then showed low growth in the following years. The progress rate of agricultural science and technology reached its peak in 2009 and 2012 with 62.70% and 72.67%, respectively, then began a slow downward trend. The contribution rate of ASTP in Guangdong Province during the 11th, 12th, and 13th Five-Year Plans were 65.09%, 65.94%, and 70.40%, respectively, showing an upward trend. Thus, the agricultural scientific and technological innovation capacity continuously improved with ASTP.

In this study, the annual contribution rate of ASTP in the Pearl River Delta, east wing, west wing, and the mountainous areas of Guangdong Province were also analyzed, as shown in Fig 2. During the 11th, 12th, and the 13th Five-Year Plans, the respective contribution rate of ASTP were as follows: 70.04%, 62.12%, and 63.00% in the Pearl River Delta; 75.71%, 61.02%, and 67.44% in the east wing; 56.89%, 62.12%, and 69.36% in the west wing; and 63.03%, 59.22% and 78.17% in the mountainous area. Except for the west wing area that showed a growing trend, all other areas showed a fluctuating trend of first decreasing and then increasing. Taking into account the topographic characteristics of various regions, the Pearl River Delta region is mostly plain, with good farming conditions, and there is a large quantity of flat arears in the west wing, with good farming conditions, however, the east wing and mountainous are covered by mountains and hills, and the farming conditions are poor, so it is difficult to implement large-scale mechanization and other modern methods. During the period of the

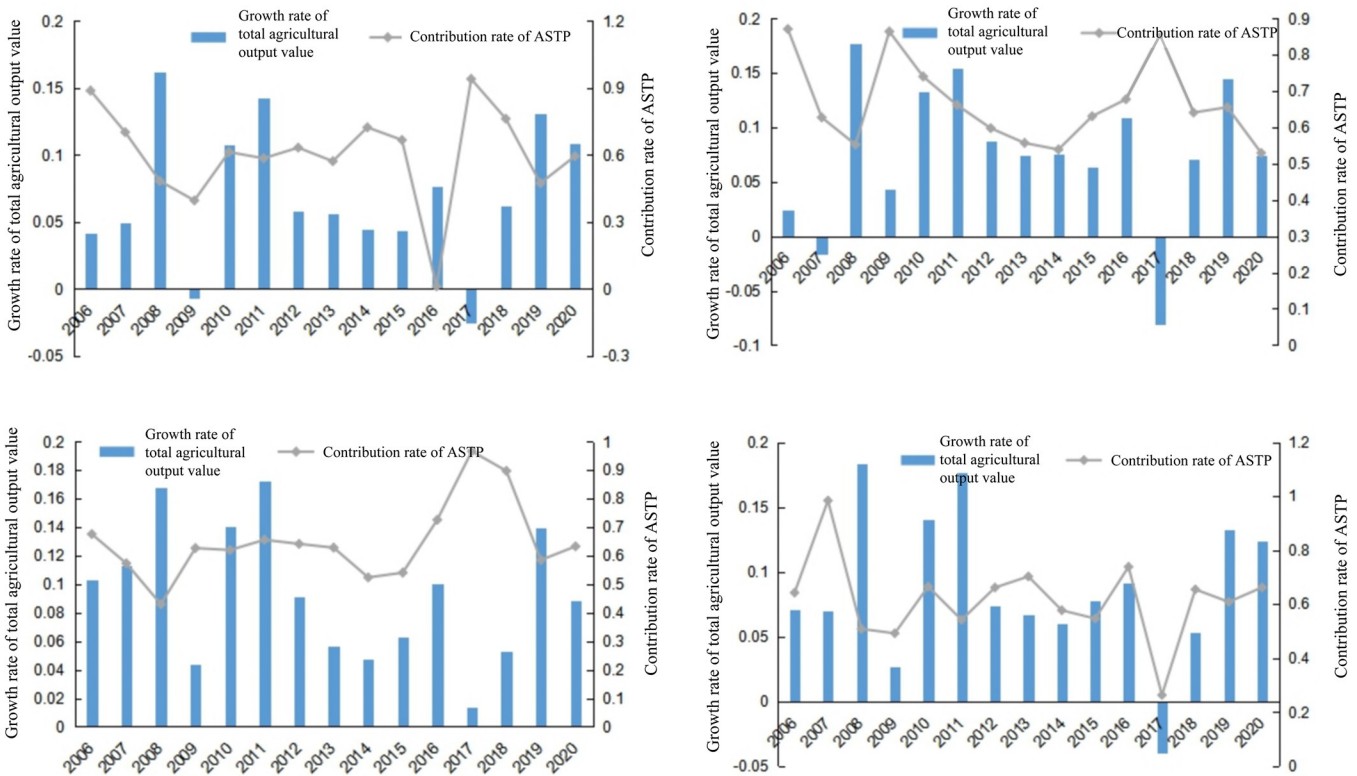

**Fig 2. Annual growth rates of total agricultural output value and contribution rate of ASTP in various areas of Guangdong Province from 2006 to 2020.**
http://stats.gd.gov.cn/gdtjnj/index.html. (a) Pearl River Delta area. (b) East wing area. (c) West wing area. (d) Mountainous area.

13th Five-Year Plan, agricultural capital investments in the Pearl River Delta and the east wing areas considerably increased. Their respective average annual growth rates of agricultural capital investments were 9.16% and 7.53% during the 13th Five-Year Plan period, an increase from the 6.18% and 5.93% of the 11th Five-Year Plan period. However, the effect of agricultural scientific and technological innovation on increasing agricultural production and income was diluted in mountainous areas. For example, the average annual growth rate of cultivated land in mountainous areas was −2.81% during the 11th Five-Year Plan period and increased to 1.44% in the 12th Five-Year Plan period. The results also show the annual growth of agricultural output values in various areas. Compared with previous years, the growth rate of agricultural output value in 2017 considerably declined. By consulting the data, Guangdong Province suffered a strong typhoon and a once-in-a-century flood in 2017, leading to direct economic losses of 31.61 billion yuan. Therefore, the ability of agricultural science and technology in Guangdong Province to resist the risk of agricultural disasters needs improvement. The growth rate of the total agricultural output value and the contribution rate of ASTP in each area fluctuated year by year, with the latter having a lag effect on the former. The lag period was 2–3 years, except in the mountainous area where this lag effect was not as apparent.

## 4.2 Correlation analysis between green agriculture and agricultural scientific and technological development based on VAR model

**4.2.1 Calculation results of carbon footprint.** Green agricultural indexes such as machinery, irrigation, chemical fertilizers, and pesticides were chosen to calculate the carbon footprint of each index in Guangdong Province from 2006 to 2020, as shown in Fig 3. From the perspective of stock index, the carbon footprint from fertilizer input is the largest, which is significantly higher than other indicators, followed by the carbon footprint from pesticides and machinery, and the carbon footprint from irrigation is the least, which indicates that in terms of agricultural production and emission reduction, the production and use of chemical fertilizers have great space for improvement. And from the perspective of incremental indicators, the results show the carbon footprints from machinery increased year by year, which is

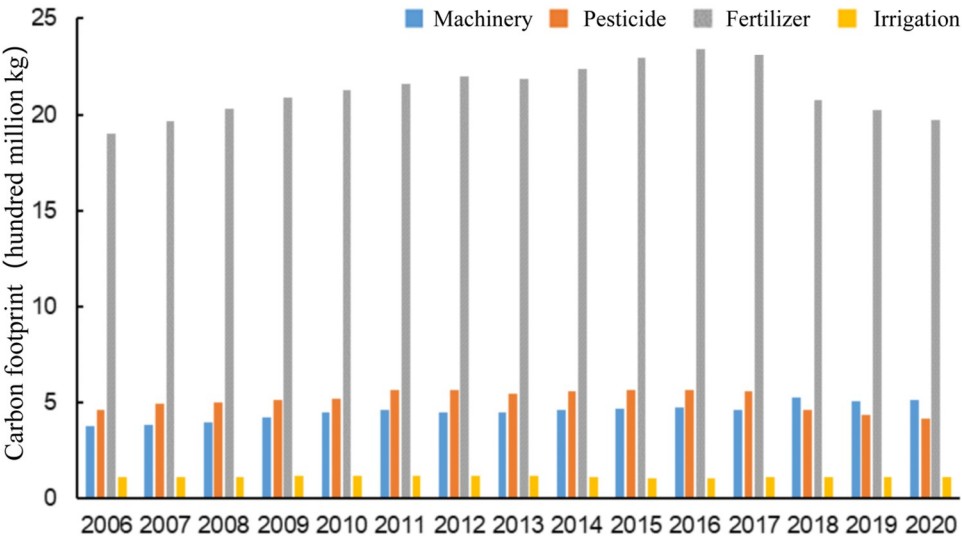

**Fig 3. Carbon footprint analysis of green agriculture indexes in Guangdong Province in different periods.** http://stats.gd.gov.cn/gdtjnj/index.html.

**Table 2. Correlation analysis of variables.** http://stats.gd.gov.cn/gdtjnj/index.html.

| Variable | Contribution rate of ASTP | Machinery | Pesticide | Fertilizer | Irrigation |
|---|---|---|---|---|---|
| Contribution rate of ASTP | 1 | | | | |
| Machinery | 0.720463 | 1 | | | |
| Pesticide | 0.350575 | −0.130970 | 1 | | |
| Fertilizer | 0.603209 | 0.355131 | 0.836942 | 1 | |
| Irrigation | 0.008992 | −0.189020 | −0.014160 | −0.271010 | 1 |

related to China's vigorous promotion of agricultural mechanization; the carbon footprints from pesicides and fertilizers increased first and then decreased; and the carbon footprints from irrigation changed little, fluctuating around 110 million kg.

**4.2.2 Correlation test of model variables.** This study included a total of 15 samples, and all variables were related to time. Their scales were reduced to mitigate the influence of variance by taking logarithms from the time series. Among the time series variables, non-stationary time series can very easily lead to "spurious regression" that may invalidate the conclusions. Therefore, the stationary test of variables must be done before the empirical analysis of the VAR model. The ADF test, which is commonly applied to the unit root test for stationarity, was selected for the analysis. After testing, the contribution rate of ASTP and the logarithms of carbon footprints of machinery, pesticides, fertilizers, and irrigation were all found as first-order stable. Table 2 shows the correlation analysis results of all variables. The results show that the contribution rate of machinery to ASTP was more significant, followed by those of chemical fertilizers, pesticides, and irrigation. Then, the Enger-Granger two-step method was used for the co-integration test to analyze the three variables—the contribution rate of ASTP, machinery, and fertilizers. First, the variables were regressed to obtain the following regression equation:

$$\ln CR = 1.03 \ln M + ECM, \tag{9}$$

where $\ln CR$ is the logarithm of the contribution rate of ASTP, $\ln M$ is the logarithm of the carbon footprint generated by machinery, and $ECM$ is the residual series. The unit root test of residual series showed that the adjoint probability was 0.018; that is, the residual series was stationary, indicating a co-integration relationship between $\ln CR$ and $\ln M$. The regression coefficient before the variable was positive, indicating that the use of machinery posed a reverse trend on agricultural scientific and technological innovation in the long term. Thus, the two variables have a long-term equilibrium relationship.

Subsequently, the Granger causality test was used on the statistical relationship between variables, as shown in Table 3 below. The results show that the utilization of pesticides and

**Table 3. Granger causality test results.** http://stats.gd.gov.cn/gdtjnj/index.html.

| Null hypothesis | F-Statistic | Prob. | Conclusion |
|---|---|---|---|
| Pesticide use is not the Granger cause for CR | 3.83383 | 0.0680 | Rejected |
| CR is not the Granger cause for pesticide use | 0.28826 | 0.7570 | Accepted |
| Fertilizer use is not the Granger cause for CR | 3.91051 | 0.0654 | Rejected |
| CR is not the Granger cause for fertilizer use | 2.58732 | 0.1360 | Accepted |
| Irrigation is not the Granger cause for CR | 0.02430 | 0.9761 | Accepted |
| CR is not the Granger cause for irrigation | 0.43047 | 0.6644 | Accepted |

Note: CR indicates the independent variable, the contribution rate of ASTP.

**Table 4. Test results of the optimal lag order.** http://stats.gd.gov.cn/gdtjnj/index.html.

| Lag | LogL | LR | FPE | AIC | SC | HQ |
|---|---|---|---|---|---|---|
| 0 | 29.59842 | NA* | 3.45E-05 | −4.59974 | −4.51892* | −4.62966 |
| 1 | 34.11565 | 6.77585 | 3.23E-05* | −4.685942* | −4.44349 | −4.77571* |
| 2 | 37.19647 | 3.59429 | 4.11E-05 | −4.53275 | −4.12866 | −4.68235 |

fertilizers was the cause for agricultural scientific and technological innovation, indicating that the change in the carbon footprints of these two variables affected the level of scientific and technological innovation applied to agriculture in Guangdong Province. However, irrigation was not the cause for such innovation, in combination with the actual industrial development, product updates and technology upgrades of pesticides and fertilizers are emerging in an endless stream, and many new products and technologies are used in production practice. However, irrigation is gradually optimized from flood irrigation to sprinkler irrigation, but the proportion of use is relatively low from the field agricultural scale.

Given that only the machinery variable was significantly correlated with the contribution rate of ASTP in the variable correlation analysis, the VAR model was used to analyze this influence relationship. Before using the VAR model, the lag order needs to be determined. The conclusions were determined using Eviews software, as seen in Table 4 below:

Given the first-order case, LR was significantly larger than that of other models whereas FPE was significantly smaller, the VAR (1) model was then established and subjected to stability test. The result is shown in Fig 4, indicating that the reciprocals of all unit roots were within the unit circle. As such, the VAR (1) model is stable and further modeling can be implemented.

The impulse response function was used to analyze the effect of the VAR (1) model on the endogenous variables and of machinery changes on ASTP, as shown in Fig 5. The influence of machinery growth on agricultural science and technology was almost zero in the initial stage, then increased in the reverse direction and became positive in the third stage. Subsequently, the influence fluctuated positively and negatively and then gradually decreased. On the whole, machinery growth had a long-term and positive influence on agricultural science and technology, meaning that the increase of diesel oil consumption promoted the agricultural transformation. In addition, for the upgrading of machinery and technology upgrading, there is also a great correlation, which directly affects the improvement of mechanical efficiency, thus saving the use of diesel fuel. Next, the influence of ASTP on machinery changes was analyzed. In the early stage, the effect was positive, then turned negative in the second stage, and fluctuated positively and negatively while approaching zero in the sixth stage. This result revealed that scientific and technological progress relatively and rapidly transformed into the agricultural field, and the effect on the use of machinery was also relatively significant and rapid. However, the short-term effect stage might be attributed to the rapid upgrading of agricultural machinery given the rapid scientific and technological development.

The mutual influence relationships between endogenous variables can be comprehended through variance decomposition. In this study, the variance decomposition of ASTP and machinery growth was carried out and the results were analyzed. Tables 5 and 6 show the contribution degree of ASTP and machinery growth, respectively. At the beginning, all the changes in ASTP came from the influence of science and technology itself. From the second stage, machinery growth influenced the ASTP, and this effect gradually increased with time. In the ninth stage, the effect became stable, maintaining an influence level of 54.5%. Therefore, ASTP required the driving force of science and technology itself, which required an increase in investment. The effect of ASTP contributed 58.69% in the early stage of machinery growth,

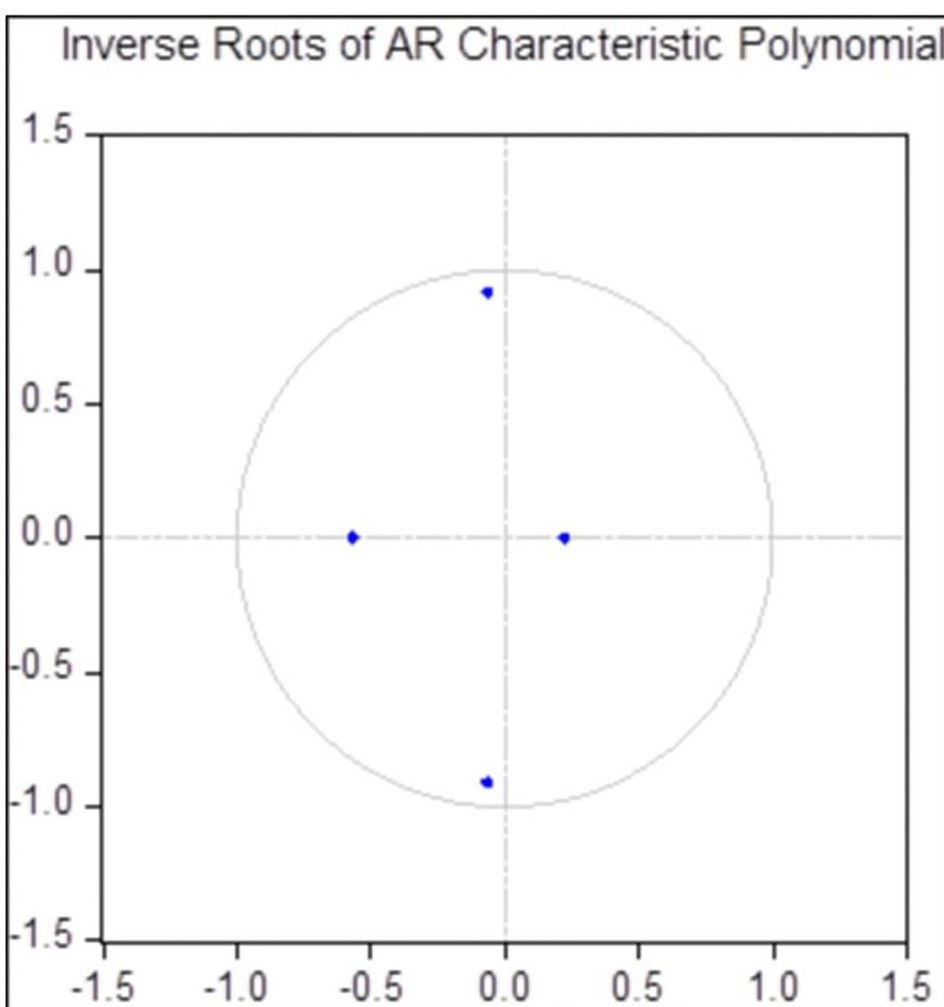

**Fig 4. AR Roots test result.** http://stats.gd.gov.cn/gdtjnj/index.html.

and then gradually decreased, and the contribution rate can reach 48.62% under their long-term influence. The scientific and technological progress still exerted considerable influence on agricultural machinery.

## 5. Conclusions and strategy suggestions

### 5.1 Conclusions

From the perspective of green agriculture, the agricultural scientific and technological innovation capacity of Guangdong Province was evaluated. Based on the statistical data of Guangdong Province from 2006 to 2020, the contribution rate of ASTP was selected as an index to evaluate the scientific and technological innovation capacity of Guangdong Province in agriculture. Meanwhile, the carbon footprints related to machinery, irrigation, chemical fertilizers, and pesticides were selected as the index related to green agriculture, and their contribution degrees to agricultural scientific and technological innovation and mutual influences were analyzed by using the VAR model. The following conclusions were drawn:

(1) On the whole, the contribution rate of ASTP in Guangdong Province showed a gradual upward trend, reaching 70.40% during the 13th Five-Year Plan period, and the agricultural

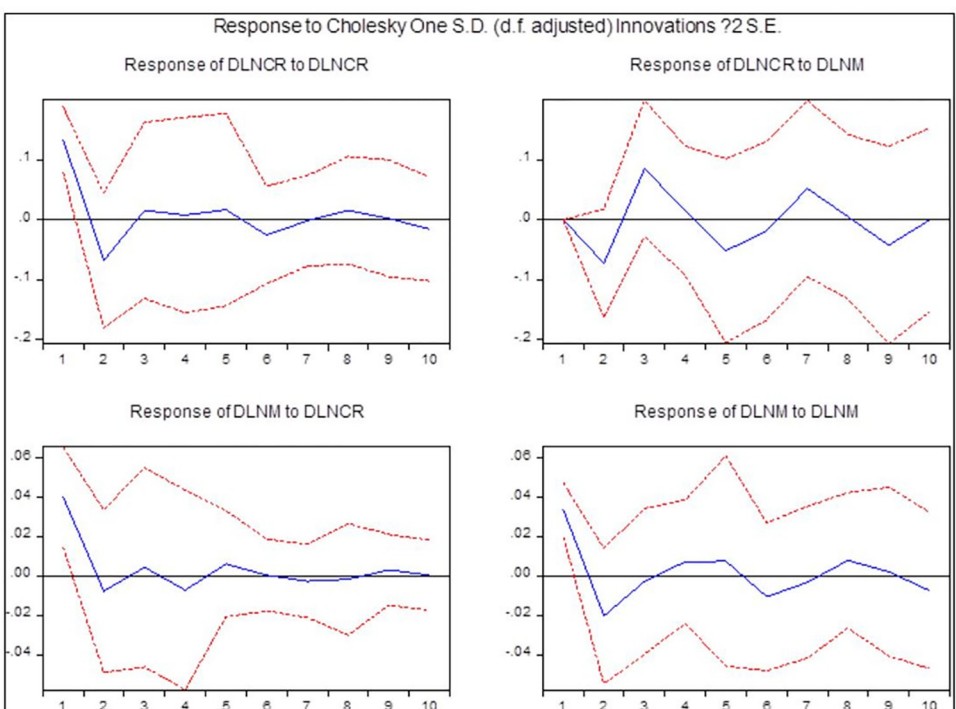

**Fig 5. Results of impulse response function based on the VAR model.** Note: DLNCR is the first-order differential contribution rate of ASTP; DLNM is the first-order differential carbon footprint of machinery. http://stats.gd.gov.cn/gdtjnj/index.html.

scientific and technological capacity continuously improved. From the regional perspective, except for the west wing, all other areas show a fluctuating trend of first decreasing and then increasing, and the effect of scientific and technological innovation on the total agricultural output value lagged, and there is a certain time difference between agricultural scientific research innovation and produce a marked effect.

(2) Analysis of the correlation between green agricultural indexes and agricultural scientific and technological innovation capacity shows that agricultural scientific and technological innovation ability is correlated with green agriculture, in which scientific and technological progress has a significant impact on machinery and fertilizer, correspondingly, machinery

**Table 5. Variance decomposition results of ASTP.** http://stats.gd.gov.cn/gdtjnj/index.html.

| Stage | S.E. | ASTP | Machinery growth |
|---|---|---|---|
| 1 | 0.133718 | 100 | 0 |
| 2 | 0.166432 | 81.13334 | 18.86666 |
| 3 | 0.187653 | 64.51949 | 35.48051 |
| 4 | 0.188416 | 64.17172 | 35.82828 |
| 5 | 0.196085 | 59.99982 | 40.00018 |
| 6 | 0.198510 | 60.14613 | 39.85387 |
| 7 | 0.205239 | 56.27309 | 43.72691 |
| 8 | 0.205914 | 56.50148 | 43.49852 |
| 9 | 0.210181 | 54.24396 | 45.75604 |
| 10 | 0.210737 | 54.48518 | 45.51482 |

**Table 6. Variance decomposition results of machinery growth.** http://stats.gd.gov.cn/gdtjnj/index.html.

| Stage | S.E. | ASTP | Machinery growth |
|---|---|---|---|
| 1 | 0.052625 | 58.68811 | 41.31189 |
| 2 | 0.056894 | 52.00160 | 47.99840 |
| 3 | 0.057130 | 52.18582 | 47.81418 |
| 4 | 0.058042 | 52.07288 | 47.92712 |
| 5 | 0.058872 | 51.74548 | 48.25452 |
| 6 | 0.059799 | 50.16071 | 49.83929 |
| 7 | 0.059934 | 50.11230 | 49.88770 |
| 8 | 0.060495 | 49.25961 | 50.74039 |
| 9 | 0.060621 | 49.33171 | 50.66829 |
| 10 | 0.061069 | 48.61725 | 51.38275 |

indexes have a significant effect on agricultural scientific and technological innovation also. Chemical fertilizers, pesticides, and irrigation have no significant effect on agricultural scientific and technological innovation. In the long term, a reverse impact trend of machinery use was observed on agricultural scientific and technological innovation, reflecting a long-term equilibrium relationship.

(3) The Granger causality test revealed that at the significance level of 10%, the use of pesticides and fertilizers was the cause for agricultural scientific and technological innovation. However, irrigation was not significant, indicating that Guangdong's scientific and technological innovation capacity in irrigation water saving need improvement.

(4) Based on the VAR model, the influence relationship between machinery and the contribution rate of ASTP was analyzed. The results show that compared with pesticide, fertilizer and other fields, scientific and technological progress rapidly and relatively transformed into the field of agricultural machinery, it is also consistent with the trend of mechanization of smallholder economy in our country. Variance decomposition also manifested that scientific and technological progress has a significant and rapid effect on agricultural machinery.

## 5.2 Policy suggestions

According to the above conclusions, Guangdong Province must gradually improve its application and transformation degree of scientific and technological innovation in the agricultural field, reduce its agricultural carbon footprint, and vigorously develop green agriculture. The following three aspects need enhancement.

(1) Enlarge the investments in agricultural scientific and technological innovation, increase the proportion of agricultural scientific research expenditure, promote the deepening of agricultural science and technology, support the R&D of more new varieties and cultivation techniques facilitating the ecological cycle and green development, promote energy conservation and carbon sequestration with the upgrading and wide application of technology, and lead a quantitative change in carbon emissions with small qualitative changes based on extensive agricultural planting.

(2) Vigorously develop new products and processes for energy conservation and emission reduction of chemical fertilizers and reduce the carbon footprint of agricultural production through scientific and technological progress in the production and use of chemical fertilizers; reasonably standardize and scientifically guide the use of agricultural products; support the popularization and application of accurate and quantitative new technologies such as soil testing and formula fertilization.

(3) Increase the R&D and popularization of agricultural machinery to achieve a more evident mutual promotion effect between agricultural machinery and ASTP, increase the R&D expenses of agricultural machinery, and promote the mechanized, intelligent, and high-efficiency development of agricultural production; develop lightweight and small-scale machinery suitable for operations in the mountains and hills according to the practical situation in Guangdong, reduce labor input, relieve the aging problem of the agricultural population, and enhance agricultural scientific and technological innovation capacity.

## 5.3 Research limitations and expectations

On the basis of previous studies, the correlation between agricultural scientific and technological innovation ability and green agriculture is further explored. However, several research limitations remain. Restricted by the availability and completeness of statistical data, for example, the data only in 15 years, namely, the 11th, 12th, and 13th Five-year Plans were used, and the time span remained to be further lengthened. Second, the measurement method for the contribution rate of ASTP that is widely used at present was proposed in 1997 and carried forward for over 20 years. Although this method remains highly recognized and widely used, the measurement factors need to be further optimized in line with the current development and scientific and technological progress, including new-type modes and methods such as socialized agricultural scientific and technological services. In future research, data on agricultural industries such as animal husbandry and aquaculture are added to further perfect the correlation between ASTP and green agriculture. Moreover, the influencing factors of this correlation require deep analysis.

## Author Contributions

**Conceptualization:** Mei Zhang, Kai Fang.

**Data curation:** Mei Zhang, Dejie Zeng.

**Formal analysis:** Mei Zhang, Danting Zhang.

**Funding acquisition:** Mei Zhang.

**Investigation:** Mei Zhang, Danting Zhang.

**Methodology:** Mei Zhang, Kai Fang.

**Project administration:** Mei Zhang.

**Resources:** Mei Zhang.

**Software:** Mei Zhang, Dejie Zeng.

**Supervision:** Mei Zhang, Danting Zhang.

**Validation:** Mei Zhang.

**Visualization:** Mei Zhang.

**Writing – original draft:** Mei Zhang.

**Writing – review & editing:** Danting Zhang.

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
