## [Decision Letter · Decision Letter 0]

30 Aug 2023

PONE-D-23-23431Relationship Between the Evaluation of Agricultural Scientific and Technological Innovation Capacity and the Influencing Factors of Green AgriculturePLOS ONE

Dear Dr. Zhang,

Thank you for submitting your manuscript to PLOS ONE. After careful consideration, we feel that it has merit but does not fully meet PLOS ONE’s publication criteria as it currently stands. Therefore, we invite you to submit a revised version of the manuscript that addresses the points raised during the review process.

We look forward to receiving your revised manuscript.

Kind regards,

László Vasa, PhD

Academic Editor

PLOS ONE

Journal Requirements:

Reviewers' comments:

Reviewer's Responses to Questions

**Comments to the Author**

1. Is the manuscript technically sound, and do the data support the conclusions?

Reviewer #1: Yes

Reviewer #2: Partly

2. Has the statistical analysis been performed appropriately and rigorously? 

Reviewer #1: Yes

Reviewer #2: Yes

3. Have the authors made all data underlying the findings in their manuscript fully available?

Reviewer #1: Yes

Reviewer #2: Yes

4. Is the manuscript presented in an intelligible fashion and written in standard English?

Reviewer #1: Yes

Reviewer #2: Yes

5. Review Comments to the Author

Reviewer #1: This paper uses the Solow growth model coupled with the Cobb–Douglas production function and vector autoregressive models to examine the driving effect of scientific and technological innovation on economic growth and green agriculture. Then it uses the contribution rate of ASTP as the assessment index to evaluate the agricultural scientific and technological innovation capacity. In this way, the paper is well organized and easy to understand. However, I still have some questions.

1. In the study of green agriculture, the carbon footprint value is taken as the measurement index, and only the stock index is analyzed. The analysis of incremental data may produce new findings and results, so it is suggested to add this part of the study.

2. The literature review cited sufficient previous studies as the basis, but some of the references were blunt, so it was too natural to highlight the problems to be explained and the proof content of the references.

3. If the data specifications in the text are inconsistent, the specifications should be unified and the accuracy should be consistent.

4. As for research methods and data, this study lacks literature annotation on the source of methods and theoretical support.

5. This paper divides Guangdong Province into four parts: the Pearl River Delta region, the East wing region, the West wing region and the mountainous region. The prefecture-level cities included in each part should be indicated in the paper.

6. Guangdong Province is mostly hilly and mountainous. For the Pearl River Delta region, the eastern region, the western region and the mountainous region, their topographic characteristics and farming conditions are quite different, which should be taken into account when analyzing the differences.

7. In the statistical relationship between variables tested by Granger causality test, changes in the carbon footprint of pesticides and fertilizers affect the degree of application of scientific and technological innovation in the agricultural field, while irrigation is not the cause of agricultural scientific and technological innovation, which indicates that the scientific and technological innovation ability of Guangdong Province in irrigation and water-saving needs to be further improved. In this paper, the analysis results of machinery, pesticides, fertilizers and irrigation should be correlated with the actual situation, and the results of data analysis should be verified by facts.

8. In the impulse response function analysis of the impact effect of endogenous variables, the correlation between diesel consumption and agricultural science and technology should take into account not only the use of machinery, but also the use of new mechanical structures and the improvement of mechanical efficiency.

Reviewer #2: According to the discussion in this paper, the author’s method is helpful. But there are still some shortcomings in the depth of the conclusion. The following are my relevant suggestions.

1. The introduction introduces the theme from the relevant situation of the current development of green agriculture and the progress of agricultural science and technology, and the respective is appropriate. The introduction content is relatively rich, however, the internal logic and relevance should be strengthened to explain the practical significance of carrying out this research.

2. In this paper, some data are not labeled with reference to sources, such as the carbon footprint coefficient of green agricultural indicators such as diesel fuel, irrigation, fertilizer and pesticide.

3. This study analyzed the influencing factors of agricultural science and technology innovation ability and green agriculture, and took the contribution rate of agricultural science and technology progress as the representative of agricultural science and technology innovation ability and the carbon footprint as the representative of green agriculture, which has a certain theoretical basis. However, in the data analysis, conclusion and strategy part of the paper, the correlation between them is not fully discussed. It proves its correlation and practical significance more deeply.

4. Broad agriculture includes agriculture (planting industry), forestry, animal husbandry, sideline industry and fishery (aquaculture industry). In this study, carbon emission indicators are mainly selected as machinery, irrigation, fertilizer and pesticide, which mainly refers to narrow agriculture and should be indicated in the paper.

5. There are many factors affecting green agriculture. This paper chooses machinery, irrigation, fertilizer and pesticides as the main indicators. Although these four are the main sources of carbon emission, there lacks basis for selection.

6. This paper takes the overall changes of the past five years as the evaluation node, corresponding to the major strategic stages of China's the 11th Five-Year Plan, the 12th Five-Year Plan and the 13th Five-Year Plan, which is highly representative. However, a brief introduction should be made to these stages to show their importance to scientific and technological progress, so as to strengthen the logic of the discussion.

7. There is a time lag between the innovation of agricultural research and its actual effect, which should be reflected in the relevant conclusions and discussions.

8. In China's actual national conditions, agricultural production is changing from intensive cultivation of small farmers to mechanization and scale, and the input of farm manure and manpower in small farmer production is huge and difficult to measure, and there are some errors in statistical data, which can be taken into account in the study.

6. PLOS authors have the option to publish the peer review history of their article (what does this mean?). If published, this will include your full peer review and any attached files.

Reviewer #1: No

Reviewer #2: No

---

## [Author Response · Author response to Decision Letter 0]

24 Sep 2023

Reviewer #1:

1. In the study of green agriculture, the carbon footprint value is taken as the measurement index, and only the stock index is analyzed. The analysis of incremental data may produce new findings and results, so it is suggested to add this part of the study.--Added.

2. The literature review cited sufficient previous studies as the basis, but some of the references were blunt, so it was too natural to highlight the problems to be explained and the proof content of the references.--Modified.

3. If the data specifications in the text are inconsistent, the specifications should be unified and the accuracy should be consistent.--Modified.

4. As for research methods and data, this study lacks literature annotation on the source of methods and theoretical support.--Added.

5. This paper divides Guangdong Province into four parts: the Pearl River Delta region, the East wing region, the West wing region and the mountainous region. The prefecture-level cities included in each part should be indicated in the paper.--Indicated.

6. Guangdong Province is mostly hilly and mountainous. For the Pearl River Delta region, the eastern region, the western region and the mountainous region, their topographic characteristics and farming conditions are quite different, which should be taken into account when analyzing the differences.--Modified.

7. In the statistical relationship between variables tested by Granger causality test, changes in the carbon footprint of pesticides and fertilizers affect the degree of application of scientific and technological innovation in the agricultural field, while irrigation is not the cause of agricultural scientific and technological innovation, which indicates that the scientific and technological innovation ability of Guangdong Province in irrigation and water-saving needs to be further improved. In this paper, the analysis results of machinery, pesticides, fertilizers and irrigation should be correlated with the actual situation, and the results of data analysis should be verified by facts.--Modified.

8. In the impulse response function analysis of the impact effect of endogenous variables, the correlation between diesel consumption and agricultural science and technology should take into account not only the use of machinery, but also the use of new mechanical structures and the improvement of mechanical efficiency.--Modified.

Reviewer #2:

1. The introduction introduces the theme from the relevant situation of the current development of green agriculture and the progress of agricultural science and technology, and the respective is appropriate. The introduction content is relatively rich, however, the internal logic and relevance should be strengthened to explain the practical significance of carrying out this research.--Modified.

2. In this paper, some data are not labeled with reference to sources, such as the carbon footprint coefficient of green agricultural indicators such as diesel fuel, irrigation, fertilizer and pesticide.--Added.

3. This study analyzed the influencing factors of agricultural science and technology innovation ability and green agriculture, and took the contribution rate of agricultural science and technology progress as the representative of agricultural science and technology innovation ability and the carbon footprint as the representative of green agriculture, which has a certain theoretical basis. However, in the data analysis, conclusion and strategy part of the paper, the correlation between them is not fully discussed. It proves its correlation and practical significance more deeply.--Modified.

4. Broad agriculture includes agriculture (planting industry), forestry, animal husbandry, sideline industry and fishery (aquaculture industry). In this study, carbon emission indicators are mainly selected as machinery, irrigation, fertilizer and pesticide, which mainly refers to narrow agriculture and should be indicated in the paper.--Indicated narrow agriculture in the paper.

5. There are many factors affecting green agriculture. This paper chooses machinery, irrigation, fertilizer and pesticides as the main indicators. Although these four are the main sources of carbon emission, there lacks basis for selection.--Modified.

6. This paper takes the overall changes of the past five years as the evaluation node, corresponding to the major strategic stages of China's the 11th Five-Year Plan, the 12th Five-Year Plan and the 13th Five-Year Plan, which is highly representative. However, a brief introduction should be made to these stages to show their importance to scientific and technological progress, so as to strengthen the logic of the discussion.--Added.

7. There is a time lag between the innovation of agricultural research and its actual effect, which should be reflected in the relevant conclusions and discussions.--Modified.

8. In China's actual national conditions, agricultural production is changing from intensive cultivation of small farmers to mechanization and scale, and the input of farm manure and manpower in small farmer production is huge and difficult to measure, and there are some errors in statistical data, which can be taken into account in the study.--Modified.

---

## [Decision Letter · Decision Letter 1]

20 Nov 2023

Relationship Between the Evaluation of Agricultural Scientific and Technological Innovation Capacity and the Influencing Factors of Green Agriculture

PONE-D-23-23431R1

Dear Dr. Zhang,

We’re pleased to inform you that your manuscript has been judged scientifically suitable for publication and will be formally accepted for publication once it meets all outstanding technical requirements.

Kind regards,

László Vasa, PhD

Academic Editor

PLOS ONE

Additional Editor Comments (optional):

Reviewers' comments:

Reviewer's Responses to Questions

**Comments to the Author**

1. If the authors have adequately addressed your comments raised in a previous round of review and you feel that this manuscript is now acceptable for publication, you may indicate that here to bypass the “Comments to the Author” section, enter your conflict of interest statement in the “Confidential to Editor” section, and submit your "Accept" recommendation.

Reviewer #1: All comments have been addressed

Reviewer #2: All comments have been addressed

2. Is the manuscript technically sound, and do the data support the conclusions?

Reviewer #1: Yes

Reviewer #2: Yes

3. Has the statistical analysis been performed appropriately and rigorously? 

Reviewer #1: Yes

Reviewer #2: Yes

4. Have the authors made all data underlying the findings in their manuscript fully available?

Reviewer #1: Yes

Reviewer #2: Yes

5. Is the manuscript presented in an intelligible fashion and written in standard English?

Reviewer #1: Yes

Reviewer #2: Yes

6. Review Comments to the Author

Reviewer #1: All comments have be addressed. The abstract and introduction of the paper is appropriate, does not require further work. The results and the discussion are well presented, no further adjustments are necessary.

Reviewer #2: All recommendations are considered. The revised version is publishable.

I wish you success in the further publishing process.

7. PLOS authors have the option to publish the peer review history of their article (what does this mean?). If published, this will include your full peer review and any attached files.

Reviewer #1: No

Reviewer #2: No

---

## [Editor Report · Acceptance letter]

22 Nov 2023

PONE-D-23-23431R1 

Relationship Between the Evaluation of Agricultural Scientific and Technological Innovation Capacity and the Influencing Factors of Green Agriculture 

Dear Dr. Zhang:

I'm pleased to inform you that your manuscript has been deemed suitable for publication in PLOS ONE. Congratulations! Your manuscript is now with our production department. 

Kind regards, 

on behalf of

Prof. Dr. László Vasa 

Academic Editor

PLOS ONE